# Hydrological modeling in glacierized catchments of Central Asia: status and challenges

Yaning Chen[1], Weihong Li[1], Gonghuan Fang[1], Zhi Li[1]

[1] State Key Laboratory of Desert and Oasis Ecology, Xinjiang Institute of Ecology and Geography, Chinese Academy of Sciences, Urumqi, 830011, China

*Correspondence to*: Yaning Chen (chenyn@ms.xjb.ac.cn)

**Abstract.** Melt water from glacierized catchments are one of the most important water supplies in Central Asia. Therefore, the effects of climate change on glaciers and snow cover will have increasingly significant consequences for runoff. Hydrological modeling has become an indispensable research approach to water resources management in large glacierized river basins, but there is a lack of focus in the modeling of glacial discharge. This paper reviews the status of hydrological modeling in glacierized catchments of Central Asia, discussing the limitations of the available models and extrapolating these to future challenges and directions. After reviewing recent efforts, we conclude that the main sources of uncertainty in assessing the regional hydrological impacts of climate change are the unreliable and incomplete datasets and the lack of understanding of the hydrological regimes of glacierized catchments of Central Asia. Runoff trends indicate a complex response of catchments to changes in climate. For future variation of water resources, it is essential to quantify the responses of hydrologic processes to both climate change and shrinking glaciers in glacierized catchments, and scientific focus should be on reducing these uncertainties.

## 1 Introduction

Climate change is widely anticipated to exacerbate water stress in Central Asia in the near future (Siegfried et al., 2012), as the vast majority of the arid lowlands in the region are highly dependent on glacier melt water supplied by the Tienshan Mountains, which are known as the 'water tower' of Central Asia (Hagg et al., 2007; Sorg et al., 2012; Lutz et al., 2014). In fact, in the alpine river basins of the northern Tienshans, glacier melt water contributes 10% of annual runoff and 20% of runoff during the drought period (Aizen et al., 1997), so climate-driven changes in glacier/snow-fed runoff regimes have significant effects on water supplies (Immerzeel et al., 2010; Kaser et al., 2010).

According to a study conducted by the Eurasian Development Bank, changes in temperature and precipitation in Central Asia have led to rapid regression in glaciers (Ibatullin et al., 2009). The overall decrease in total glacier area and mass from 1961 to 2012 to be 18±6% and 27±15%, respectively (Farinotti et al., 2015). These values correspond to a total area loss of 2,960±1,030 km$^2$, and an average glacier mass change rate of -5.4±2.8 Gt yr$^{-1}$. If the warming projections developed by the Intergovernmental Panel on Climate Change (IPCC) prove to be true, the glacierized river systems in Central Asia will

undergo unfavorable hydrological changes, e.g., altered seasonality, increased flood risk, higher and intense spring discharge and water deficiency in hot and dry summer periods, especially given the sharp rise in water demand (Hagg et al., 2006; Siegfried et al., 2012). The development of hydrological models on accounting for changes in current and future runoff is therefore crucial for water resources allocation in river basins, and includes understanding climatic variability as well as the impact of human activities on climate (Bierkens, 2015).

Hydrological modeling is an indispensable approach to water resources research and management in large river basins. Such models help researchers understand past and current changes and provide a way to explore the implications of management decisions and imposed changes. The purpose of hydrological modeling on basin scale is primarily to support decision-making for water resources management, which can be summarized as resource assessment, vulnerability assessment, impact assessment, flood risk assessment, prediction, and early warning (World Meteorological Organisation, 2009). It is important to choose the most suitable hydrological model for a particular watershed based on the area's climate, hydrology, and underlying surface conditions.

The Tienshan Mountains span several countries and sub-regions, creating a decentralized political entity of complex multi-national and multi-ethnic forms. There are three large transboundary international rivers originated in the high mountains of Central Asia. In an international river, hydrological changes are related to the interests of the abutting riparian countries (Starodubtsev and Truskavetskiy, 2011; Xie et al., 2011; Guo et al., 2015). However, as conflicts between political states may arise for any number of reasons (political, cultural, etc.), transboundary issues may result in fragmented research and thus limit the development of hydrological modeling.

Amid this potential hindrance to robust research efforts, the effect of climate change on glaciers, permafrost and snow cover is having increasing impacts on runoff in glacierized Central Asian catchments. However, solid water is seldom explicitly considered within hydrological models due to the lack of complete glacier data. Our knowledge of snow/glacier changes and their responses to climate forcing is still mostly incomplete. Analysis of current and future water resources variations in Central Asia may promote adaptation strategies to alleviate the negative impacts of expected increased variability in runoff changes resulting from climate change.

In this paper, we review hydrological modeling efforts in five major river basins originating from the Tienshan Mountains in Central Asia, namely the Tarim River Basin, the watersheds in the northern slope of the Tienshan Mountains (which includes several small river basins), the Issyk Lake Basin, the Ili River Basin, and the Amu Darya and Syr Darya Basins (Fig. 1). Their topographical characteristics, climate, vegetation together with the glacierized area are listed in Table 1. We examine the types, purpose and use of existing models and assess the constraints and gaps in knowledge. The current lack of understanding of high-altitude hydrological regimes is causing uncertainty in assessing the regional hydrological impacts of climate change (Miller et al., 2012). Snow and glacial melt as supplies of solid water are a key element in streamflow regimes (Lutz et al., 2014), so it is necessary to include glacier mass balance estimates in the model calibration procedure (Schaefli et al., 2005; Stahl et al., 2008; Konz and Seibert, 2010; Mayr et al., 2013).

## 2 Modeling hydrological responses to climate change

Changes in the amount and seasonal distribution of river runoff may have severe implications for water resources management in Central Asia. "Glacier runoff" is defined as the total runoff generated from the melting of glaciers, snow and glacier, but can also include liquid precipitation on glacierized areas (Unger-Shayesteh et al., 2013). A large number of hydrological models applied in glacierized catchments of Central Asia are basin-scale models, which contain empirical hydrological models as well as physical hydrological models (Table 2). These glacio-hydrological models are useful tools for anticipating and evaluating the impacts of climate changes in the headwater catchments of the main Asian rivers (Miller et al., 2012).

### 2.1 Current and future runoff changes

River runoff responds in a complex way to variation in climate and the cryosphere. At the same time, runoff changes also depend on dominant runoff components. Table 2 shows that annual runoff anomalies have increased to some extent (except in the western Tienshan Mountains) and inconsistencies between changes in precipitation and runoff have occurred in heavily glacierized catchments. In rivers fed by snow and glaciers, runoff has increased (e.g., in northern Tienshan Mountains) and rising temperatures dominate the runoff changes by, for instance, increasing the snow/glacier melt and decreasing snowfall fraction (ratio of solid precipitation to liquid precipitation) (Chen, 2014). Khan and Holko (2009) compared runoff changes with variations in snow cover area and snow depth. They suggested that the mismatch between decreasing trends in snow indicators and the increasing river runoff could be the result of enhanced glacier melting. Heavily glacierized river basins showed mainly positive runoff trends in the past few decades (simulated under different scenarios in the head rivers of the Tarim River Basin), while those with less or no glacierization exhibited wide variations in runoff (Duethmann et al., 2015; Kaldybayev et al., 2016).

With further warming and the resulted acceleration of glacier retreat, glacier inflection points will or have already appeared. The amount of surface water will probably decline or keep high volatility due to glacial retreat and reduced storage capacity of glaciers (Chen et al., 2015). For instance, near future runoffs are projected to increase to some extent, with increments of 13%-35% during 2011-2050 compared to 1960-2006 for the Yarkand River, -1% -18% in the 21$^{st}$ century compared to 1986-2020 under RCP4.5 for the Kaidu River, 23% in 2020 for the Hotan River (Table 2). For the long-term, however, total runoff is projected to be smaller than today. The hydrological responses to climate change around the world were discussed in Section 1.3.

### 2.2 Contribution of glacier/snow melting water in river runoff

Kemmerikh (1972) estimated the contribution of groundwater, snow and glacier melt to the total runoff of the alpine rivers in Central Asia. Based on the hydrograph separation methodology, the glacier melt contribution ranged between 5% and 40% in the plains and around 70% in upstream basins. The ratio of glacier melt contribution to runoff varies between 3.5% and

67.5% with a mean of 24.0% for the twenty-four catchments in the Tienshan Mountains based on hydrological modelling (Zhang et al., 2016a).

Distributed hydrological models provide a more useful tool for the investigation of changes in different runoff components. For example, the Variable Infiltration Capacity (VIC) model was used to calculate the components of runoff in the source river for the Tarim River. The results showed that, in terms of runoff, glacier meltwater, snowmelt water and rainfall accounted for 43.8%, 27.7% and 28.5% of the Kumalike River, and 23.0%, 26.1% and 50.9% of the Toxkan River, respectively (Zhao et al., 2013). This result is comparable to the conclusion that glacier melt accounting for 31%~36% based on isotope tracer (Sun et al., 2016). However, accurately quantifying the contributions of glacier melt, snow melt and rainfall to runoff in Central Asian streams is challenging (Unger-Shayesteh et al., 2013).

## 2.3 Glacio-hydrological responses to climate change: a comparison

To analyse the hydrological responses to climate change of the glacierized Tienshan Mountains, the responses of several major glacierized mountainous regions are discussed. For the Himalaya–Hindu Kush region, investigations suggested that a regression of the maximum spring streamflow period in the annual cycle by about 30 days, and annual runoff decreased by about 18% for the snow-fed basin, while increased by about 33% for the glacier-fed basin using the Satluj Basin as a typical region (Singh and Bengtsson, 2005). For the Tibetan Plateau, the glacier retreat could lead to expansion of lakes, e.g., glacier mass loss between 1999 and 2010 contributed to about 11.4% ~ 28.7% of the lake level rise in the three glacier-fed lakes, namely, Siling Co, Nam Co and Pung Co (Lei et al., 2013). Analysis from groundwater storage indicated that the groundwater for the major basins in the Tibetan Plateau has increased during 2003-2009 with a trend rate of +1.86 ±1.69 Gt yr$^{-1}$ for the Yangtze River Source Region, +1.14 ±1.39 Gt yr$^{-1}$ for the Yellow River Source Region (Xiang et al., 2016).

For the South American Andes, the melting at the glacier summit has been occurred. With the continually increased temperature, though glacier melt was dominated by maybe other processes in some regions, the probability seems high that the current glacier melting will continue. As the loss of glacier water, the current dry-season water resources will be heavily depleted once the glaciers have disappeared (Barnett et al., 2005).

For the Alps Mountains, many investigations have been implemented, ranging from glacier scale modelling to large basin scale or region scale modelling (Finger et al., 2015; Abbaspour et al., 2015). Glacier melt water provided about 5.28 ±0.48 km$^3$ a$^{-1}$ of freshwater during 1980–2009. About 75% of this volume occurred during July–September, providing water for large low-lying rivers including the Po, the Rhine and the Rhône (Farinotti et al., 2016). Under the context of climate change, decreases in both annual and summer runoff contributions are anticipated. For example, annual runoff contributions from presently glacierized surfaces are expected to decrease by 13% by 2070–2099 compared to 1980-2009, despite of nearly unchanged contributions from precipitation under RCP 4.5 (Farinotti et al., 2016).

For the glacierized regions, they have something in common. The annual runoff is likely to reduce in a warming climate with high spatial-temporal variation at the middle or end of the 21$^{st}$ century. Seasonally, increased snowmelt runoff and water shortage of summer runoff with the disappearing glaciers are expected. However, there are also differences in the responses

of hydrological processes to climate change. For example, the contrasting climate change impact on river flows from glacierized catchments in the Himalayan and Andes Mountains (Ragettli et al., 2016). In the Langtang catchment in Nepal, increased runoff is expected with limited shifts between seasons while for the Juncal catchment in Chile, the runoff has already been decreasing. These qualitative or quantitative differences are mainly caused by glaciation ratio, regional weather pattern and glacier property (Hagg & Braun, 2005).

However, for many glacierized catchment in the Tienshan Mountains, currently or for the next several decades, the runoff appears to be normal or even increasing trend, making an illusion of better prospect. What worth particularly mentioning is that, once the glacier storage (fossil water) melts away, the water system is likely to go from plenty to want, exacerbating water stress given the increasing water demand.

## 3 Limitations of the available hydrological models

### 3.1 Meteorological inputs in hydrological modeling and prediction

In mountainous regions of Central Asia, meteorological input uncertainty could account for over 60% of model uncertainty (Fang et al., 2015a). The greatest challenge in hydrological modeling is lack of robust and reliable complete meteorological data, especially since the collapse of the Soviet Union in the late 1980s. In this section, the value and limitations of different datasets used in hydrological modeling (e.g., station data, remote sensing data) and future prediction (e.g., outputs of GCMs, RCMs) are discussed.

3.1.1 Observational data

Traditionally, hydrological models are forced by station-scale meteorological data in or near the studied watershed (e.g., Fang et al., 2015a; Peng and Xu, 2010). However, station-scale data can only describe the climate at a specific point in space and time, and most of them located at the foot of mountains. This limitation needs to be taken into consideration when interpolating station data into basin scale under rugged terrain. Li et al. (2014) applied the interpolated gridded precipitation dataset (APPRODITE) to force the SRM model. Applying in-situ observational meteorological data is also associated with other challenges, as detailed below.

(1) Lack of stations

One of the greatest challenges inherent in station-scale meteorological data is the low density of meteorological stations. As the mountainous regions of Central Asia are characterized by complex terrain, it is inaccurate to represent the climatic conditions of basins using data from limited stations. Some researchers (Liu et al., 2016b; Fang et al., 2015a) have addressed this challenge by attempting to interpolate temperature/precipitation into a basin scale using elevation bands, based on the assumption that climate variables increase or decrease with elevation. Temperature lapse rates could also be validated using the Integrated Global Radiosonde Archive (IGRA) dataset (Li and Williams, 2008). However, this modification could not take account of the source of water vapor and mountain aspect for basins with complex landform. Due to the fact that

uniform precipitation gradients cannot be derived and temperature lapse rates are not constant throughout the year (Immerzeel et al., 2014), it is a challenge to use elevation bands to interpolate station-scale climate into basin-scale climate.

(2) Lack of homogeneity test

Most hydrological modeling studies do not factor in errors in observations, even though homogeneous climate records are required in hydrological design. In Central Asia, changes in regulation protocols or relocation of stations also lead to observational errors. Checking the input data should be the first step in hydrological modeling due to the rule of "Garbage In Yields Garbage Out".

### 3.1.2 Remote sensing data and reanalysis data

Remote sensing and reanalysis data are increasingly being used in hydrological modeling. Liu et al. (2012a; 2016b) evaluated remote sensing precipitation data of the Tropical Rainfall Measuring Mission (TRMM) and temperature data of Moderate Resolution Imaging Spectroradiometer (MODIS). The results indicated that snow storage and snowpack that were modelled using the remote sensing climate are different from those modelled using station-scale observational data. The model forced by the remote sensing data showed better performance in spring snowmelt (Liu et al., 2012a). Huang et al. (2010a) analyzed the input uncertainty of remote sensing precipitation data interpreted from FY-2. In addition to meteorological data, surface information interpreted from satellite images, e.g., soil moisture, land use and snow cover, can also be used in hydrologic modeling (Cai et al., 2014).

As demonstrated in numerous research studies, data assimilation holds considerable potential for improving hydrological predictions (Liu et al., 2012b). Cai et al. (2014) used Global Land Data Assimilation System (GLDAS) 3h air temperature data to force the MS-DTVGM model, while Duethmann et al. (2015) used the Watch Forcing Data based on ERA-40 (WFD-E40) to force the hydrological model.

Remote sensing and reanalysis data are supposed for use in large-scale hydrological modeling due to their low spatial resolution. Another limitation in using remote sensing and reanalysis data is that these data are biased to some extent. For example, the TRMM data are mostly valuable only for tropical regions, and reanalysis data, including ERA-40, NCEP/NCAR and GPCC, fail to reveal any significant correlation with station data (Sorg et al., 2012). Given the advantages and disadvantages of observation data, remote sensing data and reanalysis data, a better approach would be to combine observations and other datasets in hydrological modeling.

### 3.1.3 GCM or RCM outputs

GCMs or RCMs provide climate variables for evaluating future hydrological processes. However, the greatest challenges in applying these datasets are their low spatial resolutions (e.g., the spatial resolution of GCMs in CMIP5 ranges from 0.75° to 3.25°) and considerable biases. In addition, different GCMs or RCMs generally give different climate projections. Therefore, when forcing a hydrological model using the outputs of climate models, the evaluation results depend heavily on the selection of GCMs and consequently result in higher uncertainty in GCMs than that in other sources (e.g., scenarios, hydrological models, downscaling etc.) (Bosshard et al., 2013).

Many downscaling methods have been developed to overcome these drawbacks. Although some statistical downscaling methods such as SDSM (Wilby et al., 2002) are widely used in climate change impact studies, their use in the mountainous regions of Central Asia is rare due to the lack of fine observational data to downscale GCM outputs. To overcome the data scarcity for this region, Fang et al. (2015b) evaluated different bias correction methods in downscaling the outputs of one RCM model and used the bias-corrected climate to force a hydrological model in the data-scarce Kaidu River Basin. Liu et al. (2011) used perturbation factors to downscale the GCM outputs and force the hydrological model.

## 3.2 Glacier melt modelling

Glacier melt accounts for a large part of the discharge for the alpine basins in Central Asia as discussed above. However, most hydrological modeling does not include glacier melt and accumulation processes. For example, Liu et al. (2010) failed to account for the glacier processes in VIC model in the Tarim river; Peng and Xu (2010) missed the glacier module in Xin'anjiang and Topmodel; Fang et al. (2015a) failed to account for glacier processes though the glacier melt could contribute up to 10% of discharge of the Kaidu River Basin. Similarly, in their research on the Yarkant River Basin, Liu et al. (2016a) neglected to include the influence of glacier melt in the SWAT and MIKE-SHE model, even though the glacier covered an area of 5574 $km^2$. The most widely used hydrological models, such as the distributed SWAT, the MIKE-SHE model and the conceptual SRM model, do not as a rule calculate glacier melt processes, despite the fact that excluding the glacier processes could induce large errors in glacierized catchments. Glacier processes are complex, in that glacier melt will at first increase due to the rise in ablation and lowering of glacier elevation, and then, after reaching its peak, will decrease due to the shrinking in glacier area (Xie et al., 2006). Moreover, simulation errors can be re-categorized as precipitation or glacier melt water and consequently result in a greater uncertainty in the water balance in high mountain areas (Mayr et al., 2013).

During the last few decades, a large variety of melt models have been developed (Hock, 2005). Previous studies have investigated glacier dynamics for the mountainous regions. Among these studies, Hock (2005) reviewed glacier melt related processes at the surface-atmosphere interface ranging from simple temperature-index to sophisticated energy-balance models. Glacier models that are physical-based (e.g., mass-energy fluxes and glacier flow dynamics) depend heavily on detailed knowledge of local topography and hydrometeorological data, which are generally limited in high mountain regions (Michlmayr et al., 2008). Hence, they mostly applied to well-documented glaciers and have few applications in basin-scale hydrological models.

The temperature-index method (or its variants), which only requires temperature for meteorological input, is widely used to calculate glacier melt (Konz and Seibert, 2010). As is illustrated by Oerlemans and Reichert (2000), glaciers can be reconstructed from long-term meteorological record, e.g., summer temperature is the dominant factor for glaciers in a dry climate (e.g., Abramov glacier). In recent years, hydrologists were trying to add other meteorological variables into the calculations of glacier melt, e.g., Zhang et al. (2007b) included potential clear sky direct solar radiation in the degree-day model, and Yu et al. (2013) stated that accumulated temperature is more effective than daily average temperature for

calculating the snowmelt runoff model. Using degree-day calculation is much simpler than using energy balance approaches and could actually produce comparable or better model performance when applied in mountainous basins (Ohmura, 2001).

More recently, the melt module has been incorporated into different kinds of hydrological models. Zhao et al. (2015) integrated a degree-day glacier melt algorithm into a macroscale hydrologic model (VIC) and indicated that annual and summer runoff would decrease by 9.3% and 10.4%, respectively, for reductions in glacier areas of 13.2% in the Kumalike River Basin. Hagg et al. (2013) analyzed anticipated glacier and runoff changes in the Rukhk catchment of the upper Amu-Darya basin up to 2050 using the HBV-ETH model by including glacier and snow melt processes. Their results showed that with temperature increases of 2.2 ℃ and 3.1 ℃, the current glacier extent of 431 $km^2$ will reduce by 36% and 45%, respectively. Luo et al. (2013) taking the Manas River Basin as a case study, investigated the glacier melt processes by including the algorithm of glacier melt, sublimation/evaporation, accumulation, mass balance and retreat in a SWAT model. The results showed that glacier melt contributed 25% to streamflow, although the glacier area makes up only 14% of the catchment drainage area.

1) Paucity of glacier variation data

The existing glacier dataset, which includes the World Glacier Inventory (WGI), the Randolph Glacier Inventory (RGI), and global land ice measurements from space (GLIMS), has been developed rapidly. These data, however, generally focus on glaciers in the present time or those existing in the former Soviet Union. For example, the source data of WGI is from 1940s-1960s, and the GLMS for the Amu Darya Basin is from 1960 to 2004 (Donald et al., 2015). These data can depict the characteristics of the glacier status, but fail to reproduce glacier variation. Only a few glaciers (e.g., Abramov, Tuyuksu, Urumqi NO. 1 Glacier, etc.) have long-term variation measurements (Savoskul and Smakhtin, 2013). CAWa is intended to contribute to a reliable regional data basis of Central Asia from the monitoring stations, sampling and remote sensing. The missing glacier variation information leads to a misrepresentation of glacier dynamics.

2) Lack of glacier mass balance data

Glacier measurements reproduced by remote sensing data usually give glacier area instead of glacier water equivalent, so errors will occur when converting glacier area to glacier mass. Glaciologists normally use a specified relation (e.g., empirical) between glacier volume and glacier area to estimate glacier mass balance (Stahl et al., 2008; Luo et al., 2013). Aizen et al. (2007) applied the radio-echo sounding approach to obtain glacier ice volume. Recently, ICESat (Ice, Cloud, and land Elevation Satellite, http://icesat.gsfc.nasa.gov/) could provide multi-year elevation data needed to determine ice sheet mass balance.

This paper focuses primarily on glacier melt modules. It does not discuss snow melt processes, as hydrological models generally include them either in a degree-day approach or energy balance basis. Furthermore, this paper does not analyze water routing processes or evapotranspiration because there are several ways to simulate soil water storage change and model evapotranspiration (Bierkens, 2015).

### 3.3 Model calibration and validation

For model calibration, two important issues are discussed here: the length of the calibration period and objective functions. Generally, hydrological modeling requires several years' calibration. For example, Yang et al. (2012) indicated that a 5-year warm up is sufficient before hydrological model calibration and a 4-year calibration could obtain satisfactory model performance. More venturesomely, a 6-month calibration could lead to good model performance for an arid watershed (Sun et al., 2016). Konz and Seibert (2010) stated that one year's calibration of using glacier mass balances could effectively improve the hydrological model. Selecting the appropriate calibration period is significant, as model performance could depend on calibration data. Refsgaard (1997) used a split-sample procedure to obtain better model calibration and validation more effectively and efficiently.

Most studies on calibration procedures in hydrology have examined goodness-of-fit measures based on simulated and observed runoff. However, as the hydrological sciences develop further, multi-objective calibration is emerging as the preferred approach. It not only includes multi-site streamflow (which has proved to be advantageous compared to single-site calibration (Wang et al., 2012b), multi-metrics of streamflow (Yang et al., 2014), but also involves multiple examined hydrological components (e.g., soil moisture). Most of the studies reviewed here use the discharge to calibrate and validate the hydrological model, yet Gupta et al. (1998) argued that a strong "equifinality effect" may exist due to the compensation effect, where an underestimation of precipitation may be compensated by an overestimation of glacier melt, and vice versa. Stahl et al. (2008) suggested that observations on mass balances should be used for model calibration, as large uncertainties exist in the data-scarce alpine regions. Therefore, multi-criteria calibration and validation is necessary, especially for glacier/snow recharged regions.

Many recent studies have attempted to include mass balance data into model calibration (Stahl et al., 2008; Huss et al., 2008; Konz and Seibert, 2010; Parajuli et al., 2009). Duethmann et al. (2015) used a multi-objective optimization algorithm that included objective functions of glacier mass balance and discharge to calibrate the hydrological model WASA. Another approach for improving model efficiency is to calibrate the glacier melt processes and the precipitation dominated processes separately (Immerzeel et al., 2012b). Further, in addition to the mass balance data used to calibrate the hydrological model, the glacier area /volume scaling factor can also be calibrated with the observed glacier area change monitored by remote sensing data (Zhang et al., 2012b).

### 4 Future challenges and directions

Modeling hydrological processes and understanding hydrological changes in mountainous river basins will provide important insight into future water availability for downstream regions of the basins. In modeling the glacierized catchments of Central Asia, the greatest challenge still remains the lack of reliable and complete data, including meteorological data, glacier data, and surface conditions. This challenge is very difficult to overcome due to the inaccessibility of the terrain and

the oftentimes conflicting politics of the countries that share the region. Even so, future efforts could be focused on constructing additional stations and doing more observations (e.g., the AKSU-TARIM project; http://www.aksu-tarim.de/).

For alpine basins with scarce data, knowledge about water generation processes and the future impact of climate change on water availability is also poor. Moreover, the contribution of glacier melt varies significantly among basins and even along river channels, adding even more complexity to hydrological responses to climate change.

Uncertainty should always be analyzed and calculated in hydrological modeling, especially when evaluating climate change impact studies that contain a cascade of climate models, downscaling, bias correction and hydrological modeling whose uncertainties are currently insufficiently quantified (Johnston and Smakhtin, 2014). The evaluation contains uncertainty in each part of the cascade, such as climate modeling uncertainty, hydrological modeling uncertainty (i.e., input uncertainty, structure or modules uncertainty and parameter uncertainty), all of which could lead to a considerably wide bandwidth compared to the changes of the water resources. In contrast, by taking into account all of these uncertainties, reliable evaluation of model confidence could be acquired by decision-makers and peers.

## 4.1 Publication of model setups and input data

As was suggested by Johnston and Smakhtin (2014), the publication of model setups and input data is necessary for other researchers to replicate the modeling or build coherent nested models. From these setups and data, researchers can build their own models from existing work rather than starting from scratch. Another advantage of researchers sharing their work is to help each other evaluate existing models from other viewpoints.

## 4.2 Integration of different data sources

After appropriate preprocessing, several types of data, including remote sensing and reanalysis, could be used in hydrological modeling, as Liu et al. (2013) indicated that remote sensing data could reproduce comparable results with the traditional station data. In recent years, isotope data are increasingly used to define water components (Sun et al., 2016) and it would be a fortune for hydrologists to validate their models, or even calibrate the models (Fekete et al., 2006). The overall idea here is to build and integrate more comprehensive datasets in order to improve hydrological modeling. An example of this approach can be found in Naegeli et al. (2013), who attempted to construct a worldwide dataset of glacier thickness observations compiled entirely from a literature review.

## 4.3 Multi-objective calibration and validation

A hydrological model should not just "mimic" observed discharge but also well reproduce snow accumulation and melt dynamics or the glacier mass change (e.g. Konz and Seibert, 2010). As discussed previously, hydrological models that are calibrated based on discharge alone may be of high uncertainty and even "equifinality" for different parameters or inputs. This could happen especially when one or several modules are missing. For example, one might overestimate the mountainous precipitation or underestimate the evapotranspiration if the glacier melt module is missing. Therefore, it is

suggested to account for each hydrological component as much as possible. We strongly suggest the use of multi-objective functions and multi-metrics to calibrate and evaluate hydrological models. Compared to single objective calibration, which was dependent on the initial starting location, multi-objective calibration provides more insight into parameter sensitivity and helps to understand the conflicting characteristics of these objective functions (Yang et al., 2014). Therefore, use different

kinds of data and objective functions could improve a hydrological model and provide more realistic results.

For the data-scarce Tienshan Mountains, however, we do not recommend an over-complex or physicalized modelling of each component as lack of validation data which may result in equifinality discussed previously when the climate system stays stable. The more empirical models (enhanced temperature index approaches) could reproduce comparable results with the sophisticated, fully-physical based models (Hock, 2005). What worth mentioning is that, the physically-based glacier

models are more advancing when quantify future dynamics of glaciers and glacier/snow redistribution when the climatic and hydrologic systems are not stable (Hock, 2005). The physical models should be further developed and used in glacier modelling as long as there is enough input and validation data

Having a reliable hydrological model is important for understanding and modeling water changes, which are key issues of water resources management. The developments and associated challenges described in this paper are extrapolations of

current trends and are likely to be the focus of research in the coming decades.

**Author contribution**

Yaning Chen and Weihong Li wrote the main manuscript text; Gonghuan Fang and Zhi Li prepared Fig. 1 and gave some assistance to paper searching and reviewing. All authors reviewed the manuscript.

**Acknowledgment**

The research is supported by the National Natural Science Foundation of China (41630859; 41471030) and the CAS "Light of West China" Program (2015-XBQN-B-17).

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

**Table 1:** Summary of climatic and underlying conditions of the basins. The topography is based on SRTM data, glacier data is from RGI (Randolph glacier inventory), and climate is based on the world map of the Koppen-Geiger climate classification. Vegetation is from the land use data from Xinjiang institute of Ecology and Geography.

| Catchment | Tarim River Basin | Catchments in northern TS China | Issyk Lake Basin | Ili River Basin | Amu Darya Basin | Syr Darya Basin |
|---|---|---|---|---|---|---|
| Location | Surrounded by the Tienshan Mountains and the Kunlun Mountains | Northern Tienshan | Western Tienshan | Western Tienshan Valley | Western Tienshan and Pamir | Western Tienshan |
| **Topography** | | | | | | |
| Basin area (km$^2$) | 868,811 | 126,463 | 102,396 | 429,183 | 674,848 | 442,476 |
| Percentage of elevation > 3000m (%) | 28.00% | 13.80% | 14.50% | 4.60% | 20.50% | 9.50% |
| Glaciation area (km$^2$) | 15789 | 1795 | 994 | 2170 | 9080 | 1850 |
| **Climate** | | | | | | |
| Dominant climate | Arid cold | Arid cold | Arid cold; continental | Arid cold; continental | Arid cold; snow | Arid cold |
| **Vegetation** | | | | | | |
| Forest percent (%) | 0.7 | 10.4 | 6.4 | 4.1 | 10.9 | 2.5 |
| Pasture percent (%) | 16.7 | 14.5 | 31.2 | 28.6 | 19.4 | 17.3 |
| Percent of water, snow, ice (%) | 5.4 | 3.9 | 7.8 | 5.3 | 5.3 | 2.8 |

**Table 2:** Summary of hydrological modeling in glacierized central Asian catchments

| Catchments | Models | Major conclusions | Innovations and Limitations | References |
|---|---|---|---|---|
| **Tarim River** | | | | |
| **Tarim River Basin** | Modified two-parameter semi-distributed water balance model | Improved the original two-parameter monthly water balance model by incorporating the topographic indexes and could get comparable results with the TOPMODEL model and Xin'anjiang model. | Less input data are required; Lack of glacier and snowmelt processes. | (Peng and Xu, 2010; Chen et al., 2006) |
| | TOPMODEL model | In the Aksu River, runoff was more closely related to precipitation, while in the Hotan River, it was more closely related to temperature. | | |
| | Xin'anjiang model | Runoffs of the Aksu, Yarkand and Hotan river exhibited increasing tendencies in 2010 and 2020 under different scenarios generated from the reference years, e.g., 23% increase for the Hotan River. | | |
| | Projection Pursuit Regression (PPR) model | If temperature rises 0.5-2.0 °C, runoff will increase with temperature for the Aksu, Yarkand and Hotan River. | Lack of physical basis. | (Wu et al., 2003) |
| | VIC | For the Tarim River, runoff will decrease slightly in 2020-2025 based on VIC forced by HadCM3 under A2 and B2 when not considering glacier melt. | Lack of glacier module. | (Liu et al., 2010) |

| | | | | |
|---|---|---|---|---|
| **Tailan River** | Modified degree-day model including potential clear sky direct solar radiation coupled with a linear reservoir model | Glacier runoff increases linearly with temperature over these ranges whether or not the debris layer is taken into consideration. The glacier runoff is less sensitive to temperature change in the debris covered area than the debris free area. | Considered the effect of solar radiation and quantified the debris effect. | (Zhang et al., 2007b) |
| **Aksu River including Kumalike and Toxkan River** | Xin'anjiang model | Precipitation has a weak relationship with runoff in the Kumalike River. | Joined the snowmelt module. The model could not well capture the snowmelt/ precipitation induced peak streamflow. | (Wang et al., 2012a) |
| | VIC-3L model | Glacier melt, snowmelt and rainfall accounted for 43.8%, 27.7 and 28.5% of the discharge for the Kumalike River while 23.0%, 26.1% and 50.9% for the Toxkan River. For the Kumalike River and the Toxkan River, the runoff have increased 13.6% and 44.9% during 1970-2007, and 94.5% and 100% of the increases were attributed by precipitation increase. For the Kumalike River, glacier area will reduce by >30% resulting in decreased melt water in summer and annual discharge (about 2.8%–19.4% in the 2050s). | The model performance was obviously improved through coupling a degree-day glacier-melt scheme, but accurately estimating areal precipitation in alpine regions still remains. | (Zhao et al., 2013; Zhao et al., 2015) |

| | | | |
|---|---|---|---|
| SWIM model | The model is capable to reproduce the monthly discharge at the downstream gauge well using the local irrigation information and the observed upstream inflow discharges. About 18% of the incoming headwater resources consumed until the gauge Xidaqiao, and about 30 % additional water is consumed between Xidaqiao and Alar. Different irrigation scenarios were developed and showed that the improvement of irrigation efficiency was the most effective measure for reducing irrigation water consumption and increasing river discharge downstream. | Investigated the glacier lake outburst floods using a modeling tool. Inclusion of an irrigation module and a river transmission losses module of the SWIM model. Model uncertainties are largest in the snow and glacier melt periods. | (Huang et al., 2015; Wortmann et al., 2014) |
| WASA model | Glacier melt contributes to 35–48% and 9–24% for the Kumalike River and the Toxkan River. For the Kumalike River, glacier geometry changes lead to a reduction of 14–23% of streamflow increase compared to constant glacier geometry. | The model considered changes in glacier geometry (e.g., glacier area and surface elevation). It used a multi-objective calibration based on glacier mass balance and discharge. | (Duethmann et al., 2015) |
| The temperature and precipitation revised AR(p) model; NAM rainfall-runoff model | The AR(p) model is capable to predict the streamflow in the Aksu River Basin. | Needing less hydrological and meteorological data. | (Ouyang et al., 2007) |

| | | | | |
|---|---|---|---|---|
| **Kaidu River** | MIKE-SHE model | Compared remote sensing data and station based data in simulating the hydrological processes. Remote sensing data is comparable to conventional data. RS data could partly overcome the lack of necessary hydrological model input data in developing or remote regions. | Missing glacier melt; Lack of observation to verify the meteorological condition in the mountainous regions. | (Liu et al., 2012a; Liu et al., 2013) |
| | HBV model | When the base runoff is 100 $m^3 s^{-1}$, the critical rainfalls for primary and secondary warning flood are 50 mm and 30 mm respectively for the Kaidu River. | It underestimated the peak streamflow while overestimated the base flow. | (Fan et al., 2014) |
| | SRM including potential clear sky direct solar radiation and the effective active temperature. | Spring streamflow is projected to increase in the future based on HadCM3. | Limited observations resulted in low modeling precision. The APHRODITE precipitation performed good in hydrological modeling in the Kaidu River. | (Zhang et al., 2007a; Zhang et al., 2008; Ma et al., 2013; Li et al., 2014). |
| | SWAT | Precipitation and temperature lapse rates account for 64.0% of model uncertainty. Runoff increases -1%~18% and 4% ~20% under RCP4.5 and RCP8.5 compared to 1986-2005 based on a cascade of RCM, bias correction and SWAT model. | Quantified uncertainty resulted from the meteorological inputs. | (Fang et al., 2015a; Fang et al., 2015c) |

| | | | | |
|---|---|---|---|---|
| | Modified system dynamics model | Simulations of low-flow and normal-flow are much better than the high-flow, and spring-peak flow are better than the summer pecks in the Kaidu River. | The modified model was robust by modified snowmelt process and soil temperature for each layer to describe water movement in soil. | (Zhang et al., 2016b) |
| **Yarkand river** | MIKE-SHE model | Simulated snow pack using station data differs significantly from that using remote sensing data. | Lack of glacier moduld | (Liu et al., 2016b) |
| | Integrating Wavelet Analysis (WA) and back- propagation artificial neural network (BPANN) | Runoff presented an increasing trend similar with temperature and precipitation at the time scale of 32-years. But at the 2, 4, 8, and 16-year time-scale, runoff presented non-linear variation. | Interpreted the nonlinear characteristics of the hydro-climatic process using statistic method. | (Xu et al., 2011; Xu et al., 2014;) |
| | Degree-day model | Decreasing rate of glacier mass was 4.39 mm $a^{-1}$ resulting in a runoff increasing trend of $0.23 \times 10^8$ $m^3$ $a^{-1}$ during 1961 - 2006. Sensitivity of mass balance to temperature is 0.16 mm $a^{-1}$ $°C^{-1}$. Glacier runoff will increase 13%-35% during 2011-2050 compared to 1960-2006 with obvious increase in Summer. | The glacier dynamics are considered and the area–volume scaling factor is calibrated using remote sensing data. | (Xie et al., 2006; Zhang et al., 2012a; Zhang et al., 2012b) |
| **Tizunafu** | SRM including snow albedo | It could well simulate the runoff of the Tizinapu River. Runoff is dominated by precipitation and temperature lapse rates and snow albedo. | Lack of glacier module. | (Li and Williams, 2008) |

| | | | | |
|---|---|---|---|---|
| **Hotan River** | Integrating Wavelet Analysis (WA) and back- propagation artificial neural network (BPANN) | For the Hotan River, runoff correlates well with the 0 °C level height in summer for the north slope of Kunlun Mountains. | Interpreted the nonlinear characteristics of the hydro-climatic process. | (Xu et al., 2011) |
| **Catchments in northern slope of TS China** | | | | |
| **Manas River** | 1) SWAT model<br>2) SRM model<br>3) EasyDHM model | 1) Glacier area decreased by 11% during 1961-1999 and glacier melt contributes 25% of discharge.<br>2) Better simulation of snowmelt runoff than rainfall–runoff by the SRM. | 1) Both the glacier melt module and two-reservoir method were included in the hydrological simulations.<br>2) Snow cover calculation algorithm is added to validate model performance. | SWAT:<br>(Yu et al., 2011; Luo et al., 2012; Luo et al., 2013; Gan and Luo, 2013)<br>SRM:<br>(Yu et al., 2013)<br>EasyDHM:<br>(Xing et al., 2014) |

| | | | | |
|---|---|---|---|---|
| **Urumqi River** | 1) Isotope Hydrograph Separation (IHS) 2) water balance model 3) HBV model 4) Exponential regression 5) SRM model 6) THmodel model | 1) Glacier melt water contributes to 9% of runoff. 2) The cumulative mass balance of the glacier was -13.69 m during 1959-2008; proportion of glacier runoff increased from 62.8% to 72.1%. 3) For a glacierized catchment (glacierization ratio is 18%), the discharge will increase by 66±35% or decrease by 40 ±13% if the glacier size keeps unchanged or glacier disappears in 2041-2060. 4) Glacier runoff is critically affected by the ground temperature. | 1) The IHS method has overwhelming potential in analyzing hydrological components for ungauged watersheds. 2) Focusing on the glacierized and ablation area. 3) Considering future runoff under different glacier change scenarios. 4) This study shed light on glacier runoff estimation based on ground temperature for data scarce regions. 5) Calculated the curve of snow cover shrinkage based on MODIS data. 6) An energy balance model is proposed to close the balance equation of soil freezing and thawing. | (Kong and Pang, 2012; Sun et al., 2013; Sun et al., 2015b Chen et al., 2012; Huai et al., 2013; Mou et al., 2008; ) |

| | | | | |
|---|---|---|---|---|
| **Ebinur Lake Catchment including Jinghe River, Kuytun River and Bortala River** | 1) SWAT model and the sequential cluster method 2) Runoff Controlled Auto Regressive (CAR) model | For the Jinghe River, 85.7% of the runoff reduction is caused by human activity and 14.3% by climate change. The Jinghe River and Kuytun River exhibited a slightly increasing trend, but an adverse trend in the Bortala River. In a warm-humid scenario, runoff in the Jinghe River and Bortala River will increase while it will decrease in the Kuytun River. | Identified the effects of human activities and climate change on runoff. The CAR is based on past and present values without physical basis. | (Dong et al., 2014; Yao et al., 2014) |
| **Juntanghu Basin:** | Distributed Snowmelt Runoff Model (DHSVM) | The coupled WRF and DHSVM model could predict 24h snowmelt runoff with relative error within 15%. | MODIS snow cover and the calculated snow depth data are used in the snowmelt runoff modeling. | (Zhao et al., 2009) |
| **Issyk Lake Basin** | | | | |
| **Small rivers around the Issyk Lake** | Degree-day approach | Runoff contribution is varying in a broad range depending on the degree of glacierization in the particular sub-catchment. All rivers showed a relative increase in annual river runoff ranging between 3.2 and 36%. | The glacier melt runoff fraction at the catchment outlet can be considerably overestimated. | (Dikich and Hagg, 2003) |
| **Chu River** | SWAT-RSG model | General decrease was expected in glacier runoff (–26.6% to –1.0%), snow melt (–21.4% to +1.1%) and streamflow (–27.7% to –6.6%); Peak streamflow will be put forward for one month. | Use the glacier dynamics and assessed the model performance based on both streamflow and glacier area. | (Ma et al., 2015) |
| **Ili River Basin** | | | | |

| | | | | |
|---|---|---|---|---|
| **Gongnaisi River** | SRM model | For the runoff is sensitive to snow cover area and temperature. If temperature increases 4°C, the runoff will decrease by 9.7% with snow coverage and runoff shifting forward. | SRM is capable to model the snowmelt runoff. | (Ma and Cheng, 2003) |
| **Tekes River** | SWAT model | Glaciers retreated about 22% since 1970s, which was considerably higher than the Tianshan average (4.7%) and China average (11.5%), resulting in a decrease of proportion of precipitation recharged runoff from 9.8% in 1966-1975 to 7.8% in 2000-2008. | Using two land use data and two Chinese glacier inventories, the model could well reproduce streamflow. | (Xu et al., 2015) |
| **Ili River** | MS-DTVGM model | Daily runoff correlated closely with snowmelt, suggesting a snowmelt module is indispensable. | This method reduced dependence on conventional observation. | (Cai et al., 2014) |
| | Water Balance | Water decrease in 1911-1986 in the middle and lower reaches of the Lake Balkhash is due to decreased rainfall and reservoirs storage. | — | (Kezer and Matsuyama, 2006; Guo et al., 2011) |

**Amu Darya and Syr Darya Basin**

| | | | | |
|---|---|---|---|---|
| **Amu Darya and Syr Darya** | STREAM | The runoff of the Syr Darya declined considerably over the last 9000 years. For the Amu Darya and Syr Darya Basin, the glacier covered areas have decrease 15% and 22% in 2001-2010 compared to the baseline (1960-1990). The runoff of the Syr Darya is not so sensitive to future warming compared to the past 9000 years. For the Amu River Basin, 20-25% of the glaciers will retain under a temperature increment being 4-5°C and precipitation increase rate being 3%/°C. For the Syr Darya, runoff under the A2 and B2 scenarios will increase 3%-8% in 2010-2039, with sharpened spring peak and a slight lowered runoff from late June to August. | Simulated long term discharge for the Holocene and future period. The model includes the calculation of rainwater, snowmelt water and glacier runoff (based on the glacier altitude and equilibrium lime altitude). | (Aerts et al., 2006; Savoskul et al., 2003; Savoskul et al., 2004; Savoskul et al., 2013) |
| | AralMountain model | For the Amu Darya, glacier melt and snowmelt contribute to 38% and 26.9% of runoff, while for the Syr Darya, the proportions are 10.7% and 35.2%. Glacier will retreat by 46.4% - 59.5% by 2050 depending on selected GCM. For the Syr Darya, average water supply to the downstream will decrease by 15% for 2021-2030 and 25% for 2041-2050. For the Amu Darya the expected decreases are 13% (2021-2030) and 31% (2041-2050). | Fully simulated the hydrological processes. | (Immerzeel et al., 2012a) |

| Test sites | | Model | Findings | Remarks | Reference |
|---|---|---|---|---|---|
| Test sites "Abramov" in SyrDarya and "Oigaing" in Amu Darya | | HBV-ETH and OEZ model | Overall good model performances were achieved with the maximum discrepancy of simulated and observed monthly runoff within 20 mm. General enhanced snowmelt during spring and a higher flood risk in summer are predicted under a doubling atmospheric $CO_2$ concentration with greatest runoff increases occurring in August for the highly glaciated catchments and in June for the Nival catchment. | It considered geographical, topographical and hydrometeorological features of test sites, and reduced modeling uncertainties. This procedure requires a lot meteorological and land surface data and knowledge of the modeler. | (Hagg et al., 2007) |
| Panj River | | HBV-ETH | For the upper Panj catchment, the current glacier extent will decrease by 36% and 45%, respectively, assuming temperature increment being 2.2 °C and 3.1 °C. | Application of glacier parameterization scheme. | (Hagg et al., 2013) |
| Naryn River | | SWAT with glacier module | Glacier area has decreased 7.3% during 1973-2002. Glaciers will recede with only 8% of the small glaciers retain by 2100 under RCP8.5 and net glacier melt runoff will reach peak in about 2040 and decrease later. | Incorporated glacier dynamics and validated the model using two glacier inventories. | (Gan et al., 2015) |
| Syr Darya | | NAM model with a separate Land-ice model | Glacier volume will lose 31%±4% under SRES A2 until 2050s, and the runoff peak will shift forward by 30–60 days from the current spring/early summer towards a late winter/early spring runoff regime. | The NAM model was improved to be robust using only five freely calibrated parameters. | (Siegfried et al., 2012) |

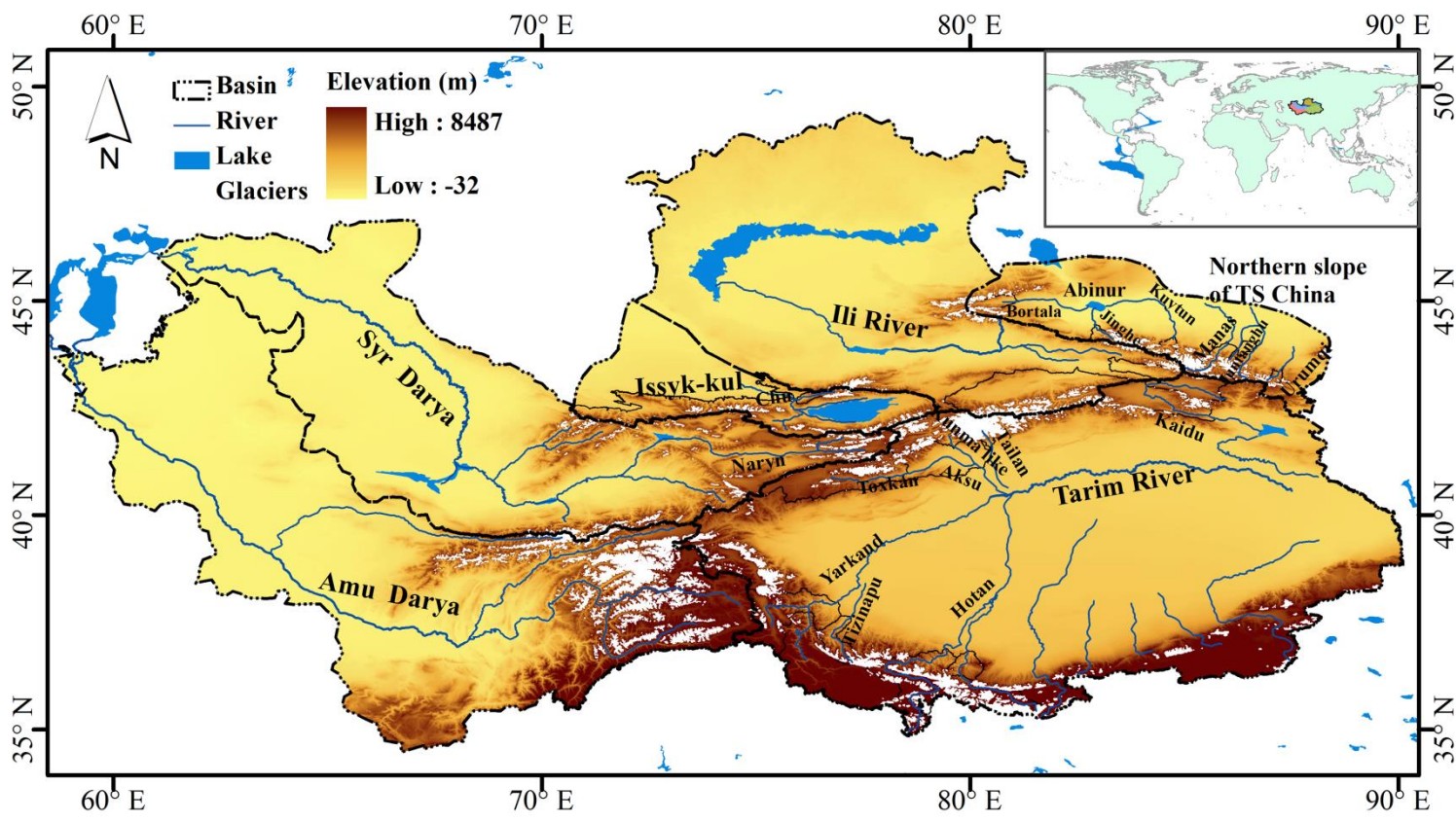

**Figure 1:** Map of Central Asian headwaters with main river basins or hydrological regions, namely the Tarim River Basin, the watersheds in the northern slope of the Tienshan Mountains, the Issyk Lake Basin, the Ili River Basin, the Amu Darya and the Syr Darya Basins. Lake outlines are from Natural Earth (http://www.naturalearthdata.com/). River system is derived based on elevations of SRTM 90 m data. Glacier information was obtained from RGI (Randolph Glacier Inventory).