# Peer review of "Hydrological modeling in glacierized catchments of Central Asia: status and challenges"

_Hydrology and Earth System Sciences, 2016_

## Referee Comment (RC1) · Anonymous Referee #1 · 29 Sep 2016

This manuscript reviews the present status, limitations and future challenges of hydrological modelling in glacierized, data-scarce Central Asia. It is a comprehensive review on recent development in hydrological modeling in glacierized catchments of Central Asia. The authors summarized hydrological responses to climate change during the past few decades and for the future period in literatures. Importantly, the limitations of hydrological modeling in the glacierized regions are summarized and discussed in terms of model inputs, model structure and model calibration. This is an interesting and important study, in particular, to discuss the specific module (glacier melt) and its limitations in hydrological modeling in Central Asia. I believe that the manuscript shed light on data-sharing and multi-objective calibration, that should be effective to help us understand the hydrological processes in data-scarce Central Asia.

I believe this manuscript has provided a comprehensive and timely review and de-

serves to be published in Hydrology and Earth System Sciences. However, the following major comments should be addressed in the further revision processes.

(1) Studies on fieldwork, e.g., glacier retreat monitoring, glacier melt modeling, should be additionally reviewed and discussed.

(2) Line 161-209: The glacier melt model (Lines 183-192) should come first in section 3.2, and then its coupling with hydrological models (for example, distributed hydrological models).

(3) If one want to quantify the hydrologic responses to climate change in central Asia (section 2.1), it would be interesting to compare researches on other parts of the world (e.g., the Alps, the Andes, the Himalayas, etc.). That would be more relevant to a general expectation in glacierized catchments.

(4) Line 82-84: need references on that.

(5) Line 85: Reorganize this sentence "glacier inflection points will or have already appeared, the amount of surface water will probably decline or remain at a high state of fluctuation".

(6) Line 100: Section 3.1 should be "Meteorological input in hydrologic .. " instead of "Input in hydrologic...". Only the meteorological inputs are discussed here and other inputs (soil, snow cover, landuse) are not well described.
* * *

---

## Referee Comment (RC2) · Y. Zhang (Referee) · 4 Oct 2016

Review on the manuscript "Hydrological modeling in glacierized catchments of Central Asia: status and challenges" No. hess-2016-325, written by Chen et al.

This manuscript comprehensively reviews hydrological modelling conducted in glacierized catchments in Central Asia. It is a very interesting synthesis focusing on the current limits, challenges and directions of hydrological modelling. The authors point it out that it is lack of glacier submodel and points directions for future hydrological modelling in those regions. The manuscript reads well. I recommend the manuscript to be accepted by HESS after a moderate to major revision is conduced. Following are my critical suggestions/comments for improving its quality 1. Glacierized catchments in Central Asia should be delineated. I cannot find spatial distribution of each catchment. 2. Need a table summarising attributes of the major catchments (i.e. Ili River, the Amu Darya, the Syr Darya, Tarim River, etc), including climate, soil, vegetation and glacier ratio, etc. 3. Table 1 should be more explicit. Please separate catchments to hydrological model. This table needs to match the table I recommend in point 2. Reviewers can then easily find catchments where how many hydrological modelling studies have been carried out in literature. 4. Table 1 can be clear by reorganisation. Following is just an example. Catchment study hydrological modelling submodels major conclusions limit 5. Need more discussion on how to improve glacier melt simulation. In section 3.2 Glacier melt, authors point that major challenge for glacier melt simulation includes lack of glacier variation data and lack of glacier mass balance data. It looks that it is a widely existed issues for hydrological modelling in high-elevation and high-latitude regions. The question is how hydrologists can improve glacier melt simulations. I think that author can have more in-depth discussion in Section 4 (i.e. if the observations incorporate into remote sensing observation to improving glacier melt observation; model paramersiation to balance equifinality and submodels; which kind of glacier models should be applied for), which will really benefit this manuscript. 6. Line 270. Semicolon should be after ' text'.

Look forward to seeing the next version of the manuscript.

Yongqiang Zhang .

---

## Author Comment (AC1) · 16 Nov 2016

**ID: HESS- 2016-325**

We would like to thank the editor's decision regarding the revision of our manuscript. We are greatly thankful for the insightful and constructive comments from the anonymous reviewer. We have carefully studied them and revised the manuscript accordingly. This document contains our specific responses to the comments.

**Responses to Anonymous Referee #1's Comments:**

This manuscript reviews the present status, limitations and future challenges of hydrological modelling in glacierized, data-scarce Central Asia. It is a comprehensive review on recent development in hydrological modeling in glacierized catchments of Central Asia. The authors summarized hydrological responses to climate change during the past few decades and for the future period in literatures. Importantly, the limitations of hydrological modeling in the glacierized regions are summarized and discussed in terms of model inputs, model structure and model calibration. This is an interesting and important study, in particular, to discuss the specific module (glacier melt) and its limitations in hydrological modeling in Central Asia. I believe that the manuscript shed light on data-sharing and multi-objective calibration that should be effective to help us understand the hydrological processes in data-scarce Central Asia. I believe this manuscript has provided a comprehensive and timely review and deserves to be published in Hydrology and Earth System Sciences. However, the following major comments should be addressed in the further revision processes.

(1) Studies on fieldwork, e.g., glacier retreat monitoring, glacier melt modeling, should be additionally reviewed and discussed.

Response: Thanks for your advice. We have added the general glacier retreat monitoring in Section 1 (Page 1 Lines 26-29) in the revised version.

The glacier melt modeling, e.g., hydrological modelling of single glacier, was additionally reviewed and added in Table 2. For example, the hydrological modelling of test site "Abramov" in Syr Darya and "Oigaing" in Amu Darya were added. This study focuses on the glacio-hydrological modeling, e.g., conclusions, limitations and directions, in the Tienshan Mountains. To make this clear, we added the explanation in our manuscript in Section 3.2.

Page 1 Lines 26-29: According to Farinotti et al. (2015), the overall decrease in total glacier area and mass from 1961 to 2012 to be $18\pm6\%$ and $27\pm15\%$, respectively. These values correspond to a total area loss of $2,960\pm1,030$ km$^2$, and an average glacier mass change rate of $5.4\pm2.8$ Gt yr$^{-1}$.

Page 8 Line 14: This paper focuses primarily on glacier melt modules. It does not include the individual glacier dynamics.

(2) Line 161-209: The glacier melt model (Lines 183-192) should come first in section 3.2, and then its coupling with hydrological models (for example, distributed hydrological models).

Response: Thanks for your comment. We have revised accordingly in the revised version.

**3.2 Glacier melt modelling**

[revised manuscript text omitted]

(3) If one want to quantify the hydrologic responses to climate change in central Asia (section 2.1), it would be interesting to compare researches on other parts of the world (e.g., the Alps, the Andes, the Himalayas, etc.). That would be more relevant to a general expectation in glacierized catchments.

Response: According to the reviewer's comment, we added Section 2.3 to compare hydrological responses for different glacierized regions worldwide including the responses of glacier runoff to climate change. Their common characteristics and differences are also addressed.

2.3 Glacio-hydrological responses to climate change: a comparison

The hydrological responses to climate change of several major glacierized mountainous regions are also discussed to make a comparison to that in the glacierized Tienshan Mountains. For the Himalaya–Hindu Kush region, investigations suggested that a regression of the maximum spring streamflow period in the annual cycle by about 30 days, and annual runoff decreased by about 18% for the snow-fed basin, while increased by about 33% for the glacier-fed basin using the Satluj Basin as a typical region (Singh et al., 2005). For the Tibetan Plateau, the glacier retreat could lead to expansion of lakes, e.g., glacier mass loss between 1999 and 2010 contributed to about 11.4% ~ 28.7% of three glacier-fed lakes, Siling Co, Nam Co and Pung Co (Lei et al., 2013). Analysis from groundwater storage indicated that the groundwater for the major basins in the Tibetan Plateau has increased during 2003-2009 with a trend rate of +1.86 ±1.69 Gt yr$^{-1}$ for the Yangtze River Source Region, +1.14 ±1.39 Gt yr$^{-1}$ for the Yellow River Source Region (Xiang et al., 2016).

For the South American Andes, the melting at the glacier summit has been occurred. With the continually increased temperature, though glacier melt was dominated by maybe other processes in some regions, the probability seems high that the current glacier melting will continue. As the loss of glacier water, the current dry-season water resources will be heavily depleted once the glaciers have disappeared (Barnett et al., 2005).

For the Alps Mountains, many investigations have been implemented, ranging from glacier scale modelling to large basin scale or region scale modelling (Finger et al., 2015; Abbaspour et al., 2015). Glacier melt water provided about 5.28±0.48 km$^3$ a$^{-1}$ of freshwater during 1980–2009. About 75% of this volume occurred during July–September, providing water for large low-lying rivers including the Po, the Rhine and the Rhône (Farinotti et al., 2016). Under the context of climate change, decreases in both annual and summer runoff contributions are anticipated. For example, annual runoff contributions from presently glacierized surfaces are expected to decrease by 16% by 2070–2099, despite of nearly unchanged contributions from precipitation under RCP 4.5 (Farinotti et al., 2016).

For the glacierized regions, they have something in common. The annual runoff is likely to reduce in a warming climate with high spatial-temporal variation at the middle or end of the 21$^{st}$ century. Seasonally, increased snowmelt runoff and water shortage of summer runoff with the disappearing glaciers are expected. However, there are also differences in the responses of hydrological processes to climate change. For example, the contrasting climate change impact on river flows from glacierized catchments in the Himalayan and Andes Mountains (Ragettli et al., 2016). In the Langtang catchment in Nepal, increased runoff is expected with limited shifts between seasons while for the Juncal catchment in Chile, the runoff has already been decreasing. These qualitative or quantitative differences are mainly caused by glaciation ratio, regional weather pattern and glacier property (Hagg & Braun, 2005).

However, for many glacierized catchment in the Tienshan Mountains, currently or for the next several decades, the runoff appears to be normal or even increasing trend, making an illusion of

better prospect. What worth particularly mentioning is that, once the glacier storage (fossil water) melts away, the water system is likely to go from plenty to want, leading to water crisis given the increasing water demand.

(4) Line 82-84: need references on that.

Response: According to the reviewer's comment, we added the reference in the revised manuscript. For example, the runoff of the Kumalike river (glaciation being 16.2%) has increased 26.2% compared to 14.9% of the Toxkan River (glaciation being 4.2%) (Duethmann et al., 2015). This is also supported by the Karatal river (Kaldybayev et al., 2016) and the Toudao Gou river.

The added reference:

Kaldybayev, A., Chen, Y., Issanova, G., Wang, H., and Mahmudova, L.: Runoff response to the glacier shrinkage in the Karatal river basin, Kazakhstan, Arabian Journal of Geosciences, 9, 1-8, doi:10.1007/s12517-015-2106-y, 2016.

(5) Line 85: Reorganize this sentence "glacier inflection points will or have already appeared, the amount of surface water will probably decline or remain at a high state of fluctuation".

Response: This sentence has been revised to "With further warming and the resulted acceleration of glacier retreat, glacier inflection points will or have already appeared. The amount of surface water will probably decline or keep high volatility due to glacial retreat and reduced storage capacity of glaciers (Chen et al., 2015)."

(6) Line 100: Section 3.1 should be "Meteorological input in hydrologic .. " instead of "Input in hydrologic: : :". Only the meteorological inputs are discussed here and other inputs (soil, snow cover, landuse) are not well described.

Response: We have updated it accordingly.

---

## Author Comment (AC2) · 17 Dec 2016

Dear Prof. Yongqiang ZHANG,

Thank you very much for your valuable comments. These helpful suggestions will help us improve this manuscript significantly. Following is the point to point reply.

This manuscript comprehensively reviews hydrological modelling conducted in glacierized catchments in Central Asia. It is a very interesting synthesis focusing on the current limits, challenges and directions of hydrological modelling. The authors point it out that it is lack of glacier submodel and points directions for future hydrological modelling in those regions. The manuscript reads well. I recommend the manuscript to be accepted by HESS after a moderate to major revision is conduced. Following are my critical suggestions/comments for improving its quality.

1. Glacierized catchments in Central Asia should be delineated. I cannot find spatial distribution of each catchment.

Response: The boundary of each catchment included in our study was added in the revised Figure 1 based on the DEM derived flow direction. Glacier information was obtained from RGI (Randolph Glacier Inventory).

[Figure]

**Figure 1:** Map of Central Asian headwaters with main river basins or hydrological regions, namely the Tarim River Basin, the watersheds in the northern slope of the Tienshan Mountains, the Issyk Lake Basin, the Ili River Basin, the Amu Darya and the Syr Darya Basins. Lake outlines are from Natural Earth (http://www.naturalearthdata.com/). River system is derived based on elevations of SRTM 90 m data. Glacier information was obtained from RGI (Randolph Glacier Inventory).

2. Need a table summarising attributes of the major catchments (i.e. Ili River, the Amu Darya, the Syr Darya, Tarim River, etc), including climate, soil, vegetation and glacier ratio, etc.

Response: Thank you for your advice. We added Table 1 summarizing the attributes of the major

catchments.

**Table 1:** Summary of climatic and underlying conditions of the basins. The topography is based on SRTM data, glacier data is from RGI (Randolph glacier inventory), and climate is based on the world map of the Koppen-Geiger climate classification. Vegetation is from the land use data from Xinjiang institute of Ecology and Geography.

| Catchment | Tarim River Basin | Catchments in northern TS China | Issyk Lake Basin | Ili River Basin | Amu Darya Basin | Syr Darya Basin |
|---|---|---|---|---|---|---|
| Location | Surrounded by the Tienshan Mountains and the Kunlun Mountains | Northern Tienshan | Western Tienshan | Western Tienshan Valley | Western Tienshan and Pamir | Western Tienshan |
| **Topography** | | | | | | |
| Basin area (km$^2$) | 868,811 | 126,463 | 102,396 | 429,183 | 674,848 | 442,476 |
| Percentage of elevation > 3000m (%) | 28.00% | 13.80% | 14.50% | 4.60% | 20.50% | 9.50% |
| Glaciation area (km$^2$) | 15789 | 1795 | 994 | 2170 | 9080 | 1850 |
| **Climate** | | | | | | |
| Dominant climate | Arid cold | Arid cold | Arid cold; continental | Arid cold; continental | Arid cold; snow | Arid cold |
| **Vegetation** | | | | | | |
| Forest percent (%) | 0.7 | 10.4 | 6.4 | 4.1 | 10.9 | 2.5 |
| Pasture percent (%) | 16.7 | 14.5 | 31.2 | 28.6 | 19.4 | 17.3 |
| Percent of water, snow, ice (%) | 5.4 | 3.9 | 7.8 | 5.3 | 5.3 | 2.8 |

3. Table 1 should be more explicit. Please separate catchments to hydrological model. This table needs to match the table I recommend in point 2. Reviewers can then easily find catchments where how many hydrological modeling studies have been carried out in literature.

Response: Thank you for your valuable suggestions. We separated the models for each catchment (including each sub-catchment) in Table 2.

[revised manuscript text omitted]

4. Table 1 can be clear by reorganisation. Following is just an example. Catchment study hydrological modelling submodels major conclusions limit.

Response: We have updated Table 1 according to your comment. Please refer to the responses to Point 3 above.

5. Need more discussion on how to improve glacier melt simulation. In section 3.2 Glacier melt, authors point that major challenge for glacier melt simulation includes lack of glacier variation data and lack of glacier mass balance data. It looks that it is a widely existed issues for hydrological modelling in high-elevation and high-latitude regions. The question is how hydrologists can improve glacier melt simulations. I think that author can have more in-depth discussion in Section 4 (i.e. if the observations incorporate into remote sensing observation to improving glacier melt observation; model parametrization to balance equifinality and submodels; which kind of glacier models should be applied for), which will really benefit this manuscript.

Response: Thanks for your comment.

In the revised manuscript, we discussed more thoroughly on how to improve the glacier melt simulation based on the present publications. First of all, Section 4.2 stressed the significance of integrated use of multiple dataset, e.g., remote sensing data, isotope tracer, observed single glacier mass balance. Section 4.3 expresses the same idea of "model parameterization". We try to emphasize the significance of multi-objective calibration and validation to handle the "equifinality" effects, especially when one or several modules are missing. In addition, some discussions are taken on the current and future development of hydrological models for the Tienshan Mountains.

**4.2 Integration of different data sources**

After appropriate preprocessing, several types of data, including remote sensing and reanalysis, could be used in hydrological modeling, as Liu et al. (2013) indicated that remote sensing data could reproduce comparable results with the traditional station data. In recent years, isotope data are increasingly used to define water components (Sun et al., 2016) and it would be a fortune for

hydrologists to validate their models, or even calibrate the models (Fekete et al., 2006). The overall idea here is to build and integrate more comprehensive datasets in order to improve hydrological modeling. An example of this approach can be found in Naegeli et al. (2013), who attempted to construct a worldwide dataset of glacier thickness observations compiled entirely from a literature review.

**4.3 Multi-objective calibration and validation**

A hydrological model should not just "mimic" observed discharge but also reproduce snow accumulation and melt dynamics or the glacier mass change (e.g. Konz and Seibert, 2010). As discussed previously, hydrological models that are calibrated based on discharge alone may be of high uncertainty and even "equifinality" for different parameters or inputs. This could happen especially when one or several modules are missing. For example, one might overestimate the mountainous precipitation or underestimate the evapotranspiration if the glacier melt module is missing. Therefore, it is suggested to account for each hydrological component as much as possible. We strongly suggest the use of multi-objective functions and multi-metrics to calibrate and evaluate hydrological models. Compared to single objective calibration, which was dependent on the initial starting location, multi-objective calibration provides more insight into parameter sensitivity and helps to understand the conflicting characteristics of these objective functions (Yang et al., 2014). Therefore, use different kinds of data and objective functions could improve a hydrological model and provide more realistic results.

For the data-scarce Tienshan Mountains, however, we do not recommend an over-complex or physicalized modelling of each component as lack of validation data which may result in equifinality discussed previously when the climate system stays stable. The more empirical models (enhanced temperature index approaches) could reproduce comparable results with the sophisticated, fully-physical based models (Hock, 2005). What worth mentioning is that, the physically-based glacier models are more advancing when quantify future dynamics of glaciers and glacier/snow redistribution when the climatic and hydrologic systems are not stable (Hock, 2005). The physical models should be further developed and used in glacier modelling as long as there is enough input and validation data

6. Line 270. Semicolon should be after ' text'.

Response: We have corrected it.